# Application of p and n-Type Silicon Nanowires as Human Respiratory Sensing Device

**DOI:** 10.3390/s23249901

**Published:** 2023-12-18

**Authors:** Elham Fakhri, Muhammad Taha Sultan, Andrei Manolescu, Snorri Ingvarsson, Halldor Gudfinnur Svavarsson

**Affiliations:** 1Department of Engineering, Reykjavik University, Menntavegur 1, 107 Reykjavik, Iceland; muhammads@ru.is (M.T.S.); manoles@ru.is (A.M.); 2Science Institute, University of Iceland, Dunhaga 3, 107 Reykjavik, Iceland; sthi@hi.is

**Keywords:** silicon nanowire arrays, metal-assisted chemical etching, sensor, humidity, breath, piezoresistance

## Abstract

Accurate and fast breath monitoring is of great importance for various healthcare applications, for example, medical diagnoses, studying sleep apnea, and early detection of physiological disorders. Devices meant for such applications tend to be uncomfortable for the subject (patient) and pricey. Therefore, there is a need for a cost-effective, lightweight, small-dimensional, and non-invasive device whose presence does not interfere with the observed signals. This paper reports on the fabrication of a highly sensitive human respiratory sensor based on silicon nanowires (SiNWs) fabricated by a top-down method of metal-assisted chemical-etching (MACE). Besides other important factors, reducing the final cost of the sensor is of paramount importance. One of the factors that increases the final price of the sensors is using gold (Au) electrodes. Herein, we investigate the sensor’s response using aluminum (Al) electrodes as a cost-effective alternative, considering the fact that the electrode’s work function is crucial in electronic device design, impacting device electronic properties and electron transport efficiency at the electrode–semiconductor interface. Therefore a comparison is made between SiNWs breath sensors made from both p-type and n-type silicon to investigate the effect of the dopant and electrode type on the SiNWs respiratory sensing functionality. A distinct directional variation was observed in the sample’s response with Au and Al electrodes. Finally, performing a qualitative study revealed that the electrical resistance across the SiNWs renders greater sensitivity to breath than to dry air pressure. No definitive research demonstrating the mechanism behind these effects exists, thus prompting our study to investigate the underlying process.

## 1. Introduction

In today’s world, the quest for a painless and simple method to monitor health parameters and detect human breath is of the greatest importance. For instance, obstructive sleep apnea syndrome is a case that necessitates precise breath monitoring during nighttime sleep. It is a sleep-related breathing disorder characterized by partial reductions (hypopnea) and complete pauses (apnea) in ventilation by pauses and reductions in airflow. It has serious consequences and the number of individuals suffering from sleep apnea complications is increasing; it has been estimated to affect up to 1 billion adults worldwide [1]. Therefore, growing research on sleep apnea demands high-quality and highly sensitive sensors [2,3]. A key to success for such an application is tracking the respiratory waveform. Also, the developing sensors are required to be able to precisely monitor the vital signs and provide alerts of possible harm in real-time. Several research studies have proposed various materials that can be used for real-time respiration monitoring, such as optical fibers, nano-structured electrochemically active aluminum, and plasma-modified graphene [4,5,6]. Despite extensive research on breath sensors, only a limited number of breath sensors have been implemented in clinical practice. The main reason is that sensors suffer from general limitations such as a high final cost and operation temperature, as well as poor sensitivity and stability [7,8,9]. In this context, semiconductor nanowires, especially SiNWs, are considered extremely promising candidates for future high-performance and highly sensitive nanoelectromechanical sensors (NEMS) [10,11,12,13]. Human breath is a complicated mixture of different gases including nitrogen, oxygen, carbon dioxide, and water vapor (humidity), that can cause slight temperature and pressure variation [14,15,16]. Meanwhile, SiNWs have exhibited a great piezoresistance (PZR) effect in a low-pressure range, which is critical for bio-compatible devices such as breath sensors [17,18,19,20]. Among the different existing approaches to synthesizing the SiNWs, the MACE method is one of the simplest methods and it offers the perspective of easy integration with microelectronics technologies at a lower cost with larger processable areas [21,22,23,24]. In this method, metal particles can be introduced through various deposition techniques, and among these, electroless deposition stands out as a straightforward method for depositing noble metals. It is typically employed when there is no stringent requirement for controlling the morphology of the resultant etched structures [25]. In a previous study, we demonstrated that SiNWs synthesized by the MACE method can detect human breath by the change in their resistance and can be used as human respiratory sensors. We also investigated the effect of incorporating germanium (Ge) nanoparticles into SiNWs (Ge: SiNWs) on the functionality of the structure as a respiratory sensor. The Ge nanoparticles were created by depositing Ge film on p-type SiNWs and annealing it at 700 °C and they were then finalized with Au electrode deposition. It was demonstrated that the inclusion of Ge nanoparticles could enhance the sensitivity and stabilize the baseline of the sensor. In contrast, thermal annealing of SiNWs samples had no discernible impact on the sensor’s response [26]. In the current paper, we forego these steps as our primary focus is the investigation of other aspects with the intent of conducting a comprehensive study to further explore and analyze the mechanism behind the sensor’s response.

Scientific research favors using Au electrodes due to their exceptional conductivity, which ensures reliable and consistent electrical connections in experiments. For commercial sensors, Au is, however, considered an expensive option, and the use of Al provides an economical alternative. The work function of a material is an important factor to consider when designing electronic devices, as it can affect the electronic properties of the device and the efficiency of electron transport across the interface between the electrode and semiconductor [27]. To investigate the electronic properties of the sensors, we selected Al as a cost-effective material of choice for this study due to its lower work function compared with Au, with silicon’s (Si) work function falling between them. The work function of Si was reported at about 4.85 eV for both intrinsic and p- or n-doped single crystalline samples [28]. In addition, the Al and Au work function was reported as 4.2 and 5.1 eV, respectively. In order to investigate the correlation among the dopant type and electrodes, four types of samples were fabricated from n-type and p-type doped Si substrates using Al and Au electrodes. Herein, we observed the resistance changes in a different direction (increase vs. decrease) upon breathing on the SiNWs with Au and Al electrodes. The direction of the resistance changes was independent of the carrier type (n or p-type). It is worth noting that there is a paucity of documented studies that report such a behavior. To uncover the underlying factors responsible for these variations, we conducted electrical characterizations of the samples. Finally, we utilized a qualitative study to probe the mechanisms behind the detectability of breath by SiNWs and the results are discussed.

## 2. Materials and Methods

A three-step MACE process was performed to synthesize random and interconnected SiNWs on p-type and n-type, single-side polished, 525 µm thick Si (001) wafers, with a resistivity ρ of 0.1–0.5 Ω cm and 0.009 Ω cm, respectively. The wafers were cut in 10 × 10 mm${}^{2}$ size samples. Theoretically, MACE of Si involves the wet etching of Si while noble metal particles or perforated films, deposited either physically or chemically onto the Si surface, are present. Subsequently, a Si substrate, partially coated with a noble metal, is immersed in an etching solution containing HF and an oxidative agent [21,23]. Generally, the Si beneath the noble metal is etched at a significantly faster rate compared with the Si without noble metal coverage. As a result, the noble metal sinks into the Si substrate, generating pores in the Si substrate or SiNWs [29]. The morphologies of the resulting etched structures would usually be defined by the configuration of the metal catalyst. This also implies that the metal layer needs to be adequately thin or porous to facilitate the movement of the liquid toward the interface. The effective removal of the dissolved material from the metal–silicon interface is also essential to facilitate rapid etching. Si etching and dissolution occur selectively only at the interface between the metal and silicon layer by the redox reaction [30]. The reduction of an oxidizing agent (H${}_{2}$O${}_{2}$ in this study) is catalyzed at the surface of noble metal nanoparticles (Ag in this study). As a result of this redox reaction, holes ($h^{+}$) are injected into the valence band of the Si substrate. The formation of these pores weakens the chemical bonds within the Si and, thus, the dissolution of the substrate specifically at the interface between the metal and the substrate [31,32]. Figure 1 depicts the MACE mechanism and formation of SiNWs. Experimentally, in step one, after cleaning the Si substrates with acetone, methanol, isopropanol, and deionized water, Ag nanoparticles were deposited on Si–wafer samples through electroless deposition, by immersing the pieces in a solution of 3 M HF and 1.5 mM AgNO${}_{3}$ for 60 s. This step created Ag nanoparticles that reside on the sample’s surface. In step two, the wires were formed by subsequently etching the samples in HF:H${}_{2}$O${}_{2}$ (5 M:0.4 M) solution for 20 min to obtain vertically aligned SiNWs. In the third step, the etching was terminated by moving the samples to de-ionized water, after which the Ag residual was removed with 60% nitric acid. A full description of the synthesis process can be found in a previous publication [33]. The top-view and cross-section images of the samples were investigated using a scanning electron microscope (SEM, Zeiss Supra 35), which is presented in Figure 2. The top-view image of SiNWs reveals a 3D structure of interconnected wires forming a rigid network. It also reveals that the synthesized SiNWs form bundles at the tips of the wires as a consequence of capillary forces being activated when extracting the samples out of the etching solution. In the cross-sectional image, a uniform etching depth of around 6 µm, is evidenced.

Four types of samples were prepared and investigated: p-type SiNWs with Al and Au electrodes, and n-type SiNWs with Al and Au electrodes. For electrical measurements, 150 nm thick metal electrodes were deposited in a co-planar configuration on the NWs samples. A schematic of the co-planar configuration is shown in Figure 1. The Au and Al electrodes were deposited by an electron-beam evaporator (Polyteknik Cryofox Explorer 600 LT). For all samples, the distance between the two co-planar electrodes was kept constant at 6 mm by using a hard mask. The electrical characteristics of the samples were analyzed by current–voltage (I–V) measurements using a Keithley 2400 source measure uni.

Human respiratory sensing was investigated using all of the fabricated SiNWs samples. The sensor samples were each mounted on a thin sample holder. The sensor was mounted firmly onto the philtrum of a volunteer, using double-sided tape. Thin copper wires were used to connect the electrodes on the samples to the board. Figure 3 provides a schematic of the measurement setup. A 3D schematic of the sensor is magnified along with an inset of an example of a long cycle breathing of a volunteer. It is worth noting that in the inset graph, the abnormal breathing pattern at 200 s was due to the coughing of the volunteer. A more detailed discussion is provided in the respiratory sensing subsection. The connection between the signal processing unit and the sensor was properly dressed within the cannula tube to avoid loose connections and/or interference with unintended objects. A Keithley 2400 source measure unit was used to apply a constant bias voltage of 2 V, and to monitor and record the corresponding resistance. All experiments were performed at room temperature to reduce random errors.

## 3. Results and Discussion

### 3.1. Electrical Characterization

The work-function of a material can affect the electronic properties of the device and the efficiency of electron transport across the interfaces between the electrodes and the semiconductor element(s). The work-function of semiconductors directly depends on the position of the Fermi level, which in turn depends on the density of states, temperature, carrier density, and doping concentration [34]. The work-function of Si is reported at about 4.85 eV for both intrinsic and p- or n-doped single crystalline samples [28]. In an experimental study by Novikov et al. [35], the dependence of the work-function of silicon to doping concentration was shown to be in the range of only ±0.03 eV for an order of magnitude different doping concentration. Whereas in our study, the resistivity of the n-type SiNWs samples was an order of magnitude lower than that of the p-type SiNWs samples, the work-function was expected to be just slightly different for both samples. The two metals used as electrodes, Au and Al, had work-functions whose levels were on different sides of silicon’s work-function, or 4.2 and 5.1 eV, respectively. The I–V curves of the SiNWs structures with co-planar electrodes were measured for all the samples listed in Table 1. The shape of the I–V curve of a semiconductor depends on the interplay between the work-functions of the electrodes (typically metals) ($\phi_{m}$) and the semiconductor ($\phi_{s}$), and on the carrier type and its concentration [36]. When a metal and a semiconductor are in contact with each other, free electrons will transfer between them due to the work-function’s difference. Typically, when $\phi_{m}<\phi_{s}$ in an n-type semiconductor, there is no Schottky barrier and the metal–semiconductor contact is ohmic, and when $\phi_{m}>\phi_{s}$ barrier forms at the metal–semiconductor interface, which lead to Schottky contact. An opposite trend occurs in p-type semiconductors [36].

Using Au and Al as metallic electrodes on n- and p-type SiNWs, ohmic and Schottky contacts were obtained as shown in Figure 4. For p-type SiNWs, an ohmic behavior was obtained with Au electrodes, Au/p-SiNWs/Au (as $\phi_{Si}<\phi_{Au}$), and Schottky contact with Al electrodes, Al/p-SiNWs/Al (as $\phi_{Si}>\phi_{Al}$). In contrast, for n-type SiNWs, a Schottky behavior was obtained with Au electrodes (Au/n-SiNWs/Au), and a semi-ohmic behavior was observed with Al electrodes (Al/n-SiNWs/Al).

### 3.2. Respiratory Sensing

The breathing patterns were obtained by exhaling and inhaling over the samples for three different breathing modes, i.e., normal (NB), rapid (RB), and deep breathing (DB). Two sequences were repeated for each mode to show the repeatability of the response and possible threat time according to the stop or pause in breathing. The highlighted gray color in the following graphs shows the pause in breathing or apnea. The respiratory tests for the same SiNWs with different electrodes showed similar responses regarding the breathing waveform response time (roughly measured as 0.1 s). All our sensors could detect different breathing modes with changes in frequency (for normal, rapid, and deep breathing modes), as the resistance variation and the response time harmonized with the breath frequency. Also, no significant difference between n-type and p-type SiNWs responses was observed. The main observed difference was in the direction of resistance changes for different electrodes, whether it increased or decreased upon exhaling. As seen in Figure 5 and Figure 6, the sensor’s resistance increased with exhaling and decreased with inhaling for both n and p-types samples with Au electrodes ($\phi_{Si}<\phi_{Au}$). The opposite behavior, thus decreasing resistance upon exhaling and increasing resistance upon inhaling, occurred when the Al electrodes were replaced with the higher work function Au, as illustrated in Figure 7 and Figure 8 (as $\phi_{Si}>\phi_{Al}$).

We define the sensitivity *S* of the SiNWs sensors according to the rate of change of resistance:(1)$$S=\frac{\Delta R}{R}=\frac{R-R_{0}}{R_{0}}\times 100\%,$$
where *R* is the resistance value of the sensor after breathing (exhaling), and $R_{0}$ is the initial value of the sensors under fixed voltage bias. In order to obtain the sensitivity of each sensor, the same measurements were repeated 10 times on each sample. Figure 9 indicates the sensitivity of sensors obtained by Equation (Equation 1). The error bars on each graph bar indicate the range of the sample’s responses for 10 individual measurements. The negative sensitivity was attributed to the presence of Al electrodes, where the resistance exhibited a decrease during the act of breathing, in contrast with the behavior of the Au electrodes. It is seen that n-type SiNWs with Au electrodes, yielding Schottky contacts, demonstrated the highest level of sensitivity. This finding aligns with numerous studies that highlight the significance of Schottky barriers in the fabrication of highly sensitive semiconductor sensors [37,38,39,40,41]. The main reason may stem from the fact that the current passing through the Schottky barrier formed at the contact area was dominantly controlled by the barrier characteristic, which was very sensitive to the environment around this small area, such as molecule adsorption and fluctuations in humidity.

We performed a comparison test between the breath sensor of SiNWs and a commercial respiratory sensor made by Nox Medical company. Nox A1s™ PSG System is an expensive commercialized sensor that is currently being used in studying sleep apnea or performing other sleep studies [42]. This particular sensor quantifies the nasal airflow pressure and lacks the capability to discern mouth breathing, which is prevalent in sleep apnea. Figure 10 shows the comparison of simultaneous breathing signals recorded by the SiNWs sensor (n-SiNWs/Au) and Nox Medical sensor. The intensity values were normalized in order to ease the comparison. A good correlation could be observed among the responses of the sensors. Nevertheless, SiNWs sensor’s response appeared to exhibit a heightened sensitivity when discerning finer details. This observation may be attributed to the gas-sensing properties of SiNWs. Human breath is inclusive of a diverse variety of gases and volatile organic compounds (VOCs) [43,44,45]. It is plausible that SiNWs have the capability to detect these components within the breath. The detailed waveform observed in breathing patterns may be linked to the presence and fluctuations of these VOCs that can be the footprint of a specific health state. However, conducting a profound investigation in this regard necessitates a comprehensive study, which is beyond the scope of the present study. Finally, sensors were tested for durability after being aged for more than six months in an ambient atmosphere. The samples were tested for longer and repeated breathing cycles, which are presented in Figure 3 as an instant for Au/p-SiNWs/Au sample. No feature loss was evident, indicating a good repeatability and endurance level of the SiNWs. For a long term stability in ambient atmosphere and daily usage, such a structure renders a superior candidate to sensors reported in various studies [38,46,47,48].

### 3.3. Respiratory Sensing Mechanism

The fundamental mechanism responsible for the detectability of breath by SiNWs lacks documented evidence and is under debate. Gosh et al. [49] fabricated a breath sensor from arrays of periodic n-type silicon nanorods [49] and attributed its function to the PZR effect. According to the model of the PZR effect, any pressure applications such as airflow on the SiNWs surface can impart sufficient normal pressures that can deform the SiNWs and cause a relative change in the resistance [17]. Controversially, a giant longitudinal PZR effect was observed on p-type SiNWs and was attributed to surface states [18]. At the same time, the PZR coefficient of n-type silicon nanowires has been reported to be of the same order of magnitude as its bulk counterpart under large strain [50]. Our previous study showed that hydrogenation on p-type SiNWs dramatically decreased its sensitivity. As hydrogenation is known to passivate shallow and deep defects in both n and p-type Si, these results strongly indicate that surface states play a significant role in the PZR effect [17]. Surface states of the SiNWs may also play an important role in the humidity-sensing mechanism described in a review article by Akbari-Saatlu et al. for metal-oxide semiconductors [51]. The humidity sensing mechanism is well-known to be based on capacitance/resistance variations due to the adsorption and desorption of water molecules on SiNWs [52,53]. It is documented that the water adsorption on nanowires changes the conductance and the capacitance of porous silicon as a result of changes in the dielectric constant or changes in its electrical resistance, and possible chemisorption and/or physisorption on its surface [54]. The water molecule (humidity) has a large electric dipole moment, 1.84 Debye. When water molecules approach the surface of SiNWs, an electrostatic capacitance is created, which has been reported in previous experimental work, and the capacitance measurements could be related to the relative humidity [55]. In our case, due to this capacitive effect, a fraction of charge carriers alters at the surface of SiNWs in the presence of humidity, consequently changing the conductance or the resistance of the device, when the relative humidity fluctuates constantly during respiration. Such changes can be detected by measuring the current through the sensor under a constant applied bias voltage. In order to demonstrate the effect of humidity on the response of SiNWs to human breath, we performed a qualitative study using normal air and N${}_{2}$ gas on the SiNWs sensor. Samples were alternatively exposed to a compressed air outlet and N${}_{2}$ gas. The outlet pressure for compressed air and the distance between the outlet and the sample were carefully adjusted to mimic the conditions where the sample was placed in front of the nostril for breath sensing. The humidity level changes were measured by a commercial humidity sensor (SparkFun Humidity Sensor-HTU21D) in real-time with (a) air, (b) N${}_{2}$ gas, and (c) breath exposure. Figure 11 shows the measured humidity changes at the SiNWs. Its level decreased to roughly 10% and 0%, from a room humidity of 20% for compressed air and N${}_{2}$, respectively, and increased to almost 60% due to human breath. Figure 12 shows the response of SiNWs with Au electrodes to the mentioned test. It was observed that the SiNWs resistance increased upon exposure to compressed air, N${}_{2}$ gas, and human breath, with a sharper and higher response to breath. The increased sensitivity to breath, as opposed to dry air, suggests the potential of SiNWs for humidity-sensing applications. Resistance increments upon different humidity variations indicate that the observed response was a combined result of both pressure and humidity variation. This result was also in good agreement with our previous study on the PZR effect on SiNWs [17]. We investigated the iso-static pressure variation effect on SiNWs and observed a dramatic increase in resistance by more than two orders of magnitude when pumping the air out of the vacuum chamber and decreasing the humidity and pressure. Thus, it can be concluded that the breath-sensing mechanism was a combination of the piezoresistive response and humidity-sensing effect.

## 4. Conclusions

In conclusion, we presented highly sensitive human respiratory sensors based on SiNWs synthesized through a metal-assisted chemical etching method. These sensors possess great potential for various healthcare applications, such as medical diagnoses and monitoring sleep apnea. A critical aspect of our research was the exploration of the carrier type and electrode’s effect on the functionality of the sensor. The choice of electrode material impacts the electronic properties and electron transport efficiency at the electrode–semiconductor interface, and we demonstrated the subsequent effects. The results of this investigation revealed a distinct directional variation in the sensors’ response when using Au and Al electrodes. This phenomenon can be attributed to the differences in work functions between the metal electrodes and SiNWs, with Au having a higher work function and Al having a lower work function in comparison with the Si work function. The respiratory sensing results indicate that n-type SiNWs with Au electrodes, yielding Schottky contacts, demonstrated the highest level of sensitivity. This aligns with previous research in the field that highlighted the importance of Schottky barriers in fabricating highly sensitive semiconductor sensors. One particular aim of the study was to determine whether the changes in the resistance could be explained based on the mechanical deformation (thus, PZR) of the SiNWs or on changes in humidity. A qualitative study using compressed air and N${}_{2}$ gas (dry air pressure) on the SiNWs sensors revealed a response similar to breath exposure on the SiNWs, but less intense. It was therefore concluded that the origin of the device’s sensitivity was a combination of PZR and a humidity effect. Additionally, a comparative analysis with a commercial respiratory sensor was performed, which indicates the SiNWs sensor’s increased sensitivity in discerning finer details. SiNWs are well-documented as being capable of detecting various gases and volatile organic compounds that can be presented in breath, offering a potential fingerprint of specific health states. This discovery, although promising, demands further in-depth exploration. Lastly, the significant durability and stability of the SiNWs sensors, even after being aged in an ambient atmosphere for over six months, make them strong candidates for long-term and daily usage in healthcare applications.

## Figures and Tables

**Figure 1 sensors-23-09901-f001:**
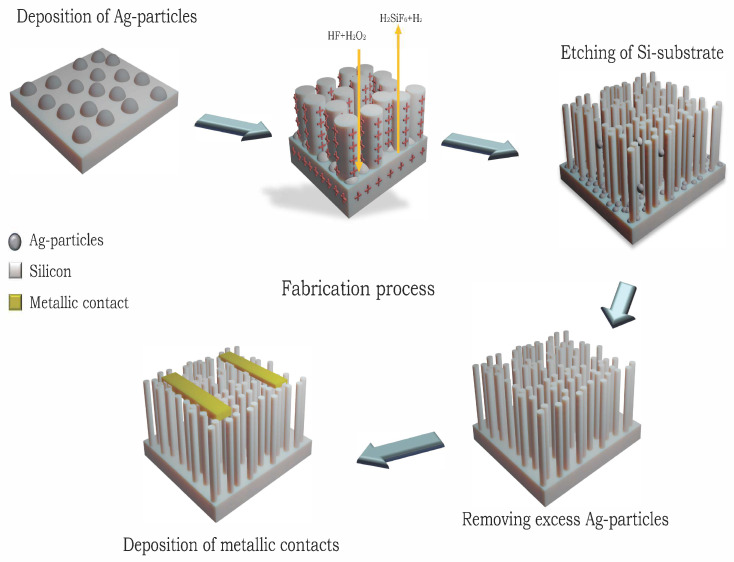
Schematic of the fabrication process of SiNWs using MACE method.

**Figure 2 sensors-23-09901-f002:**
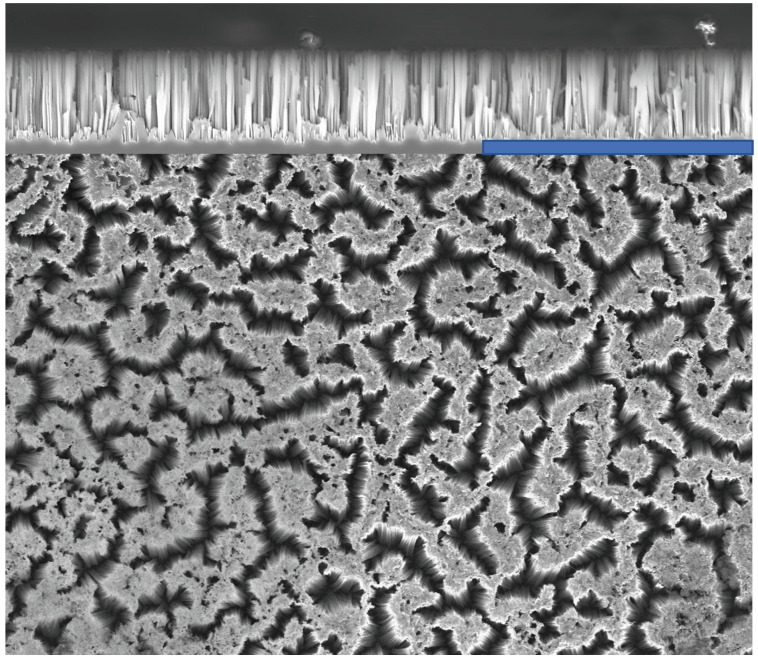
Cross-sectional and top-view SEM micrograph of n-type SiNWs obtained by MACE. The blue-colored scale bar provided is 20 µm.

**Figure 3 sensors-23-09901-f003:**
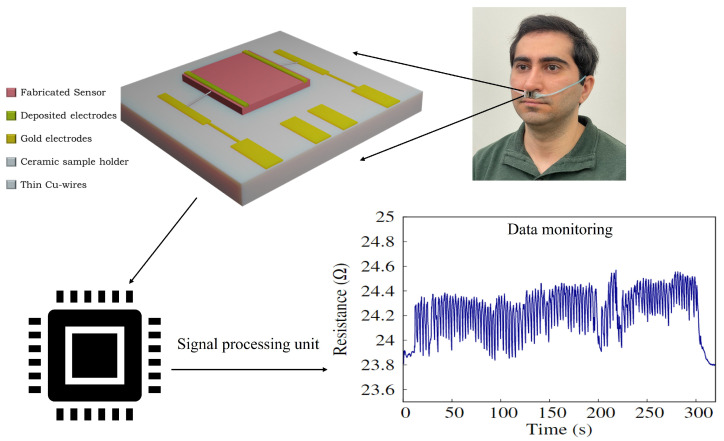
Schematic of measurement setup for breath monitoring; inset as an example of long cycle breathing in NB mode on the Au/p-SiNWs/Au sample after aging it for 6 months.

**Figure 4 sensors-23-09901-f004:**
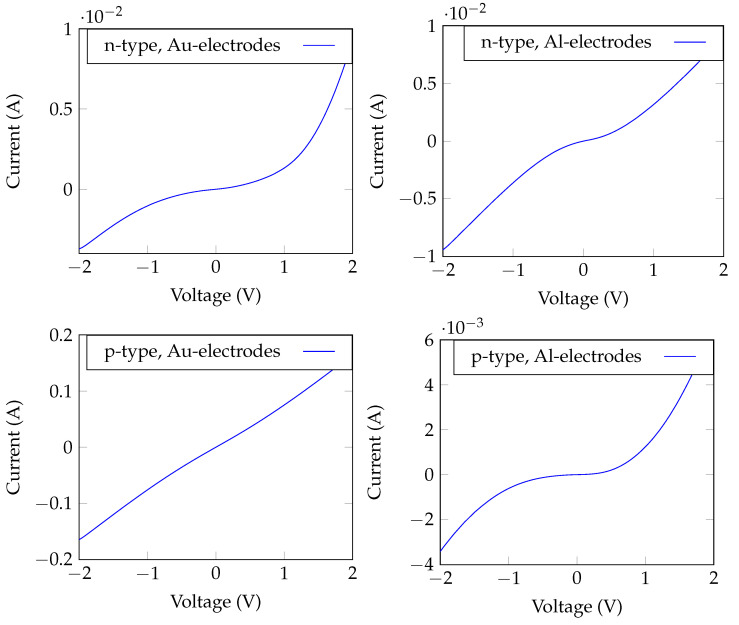
I–V characteristics of the four types of fabricated samples (p-type and n-type SiNWs with Al and Au electrodes).

**Figure 5 sensors-23-09901-f005:**
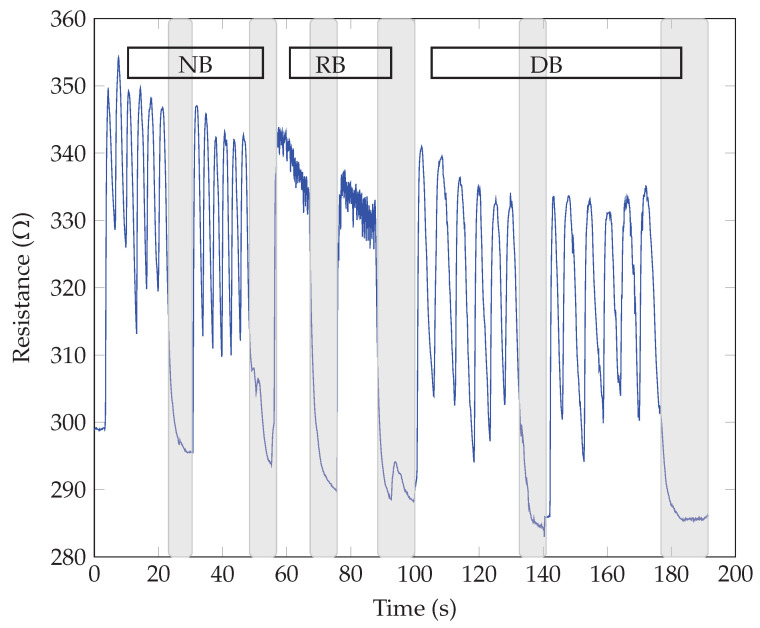
Breath sensing test of p-type SiNWs with Au electrodes.

**Figure 6 sensors-23-09901-f006:**
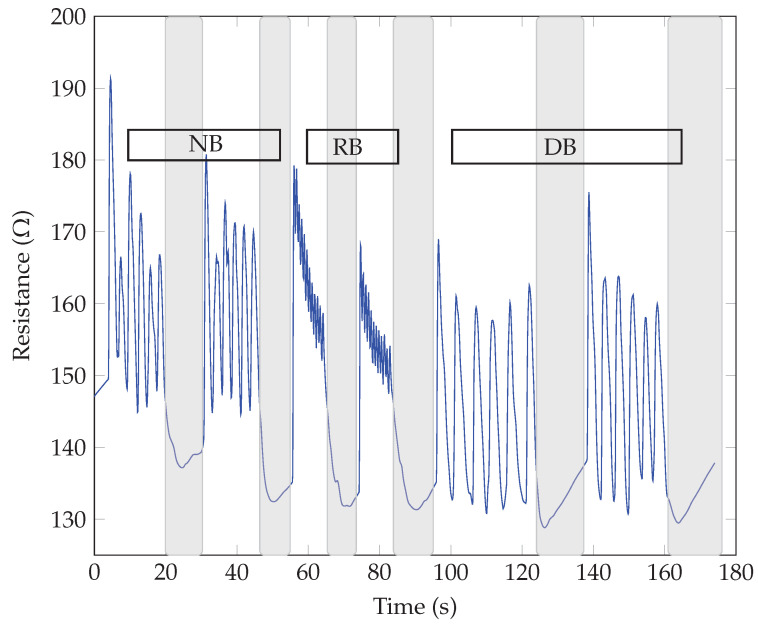
Breath sensing test of n-type SiNWs with Au electrodes.

**Figure 7 sensors-23-09901-f007:**
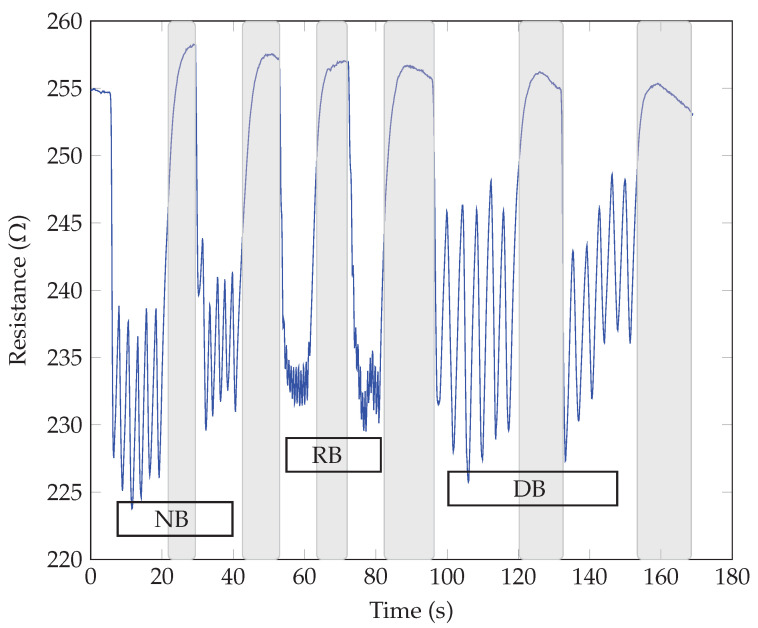
Breath sensing test of p-type SiNWs with Al electrodes.

**Figure 8 sensors-23-09901-f008:**
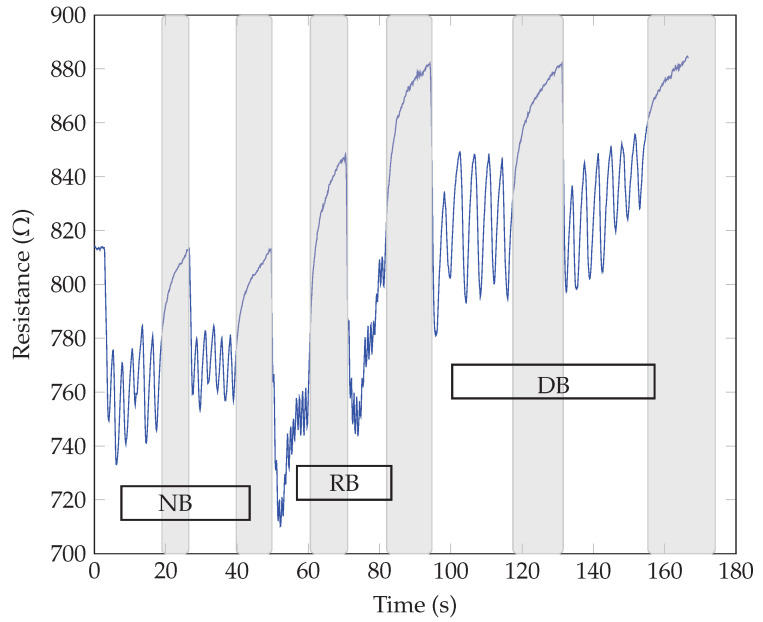
Breath sensing test of n-type SiNWs with Al-electrodes.

**Figure 9 sensors-23-09901-f009:**
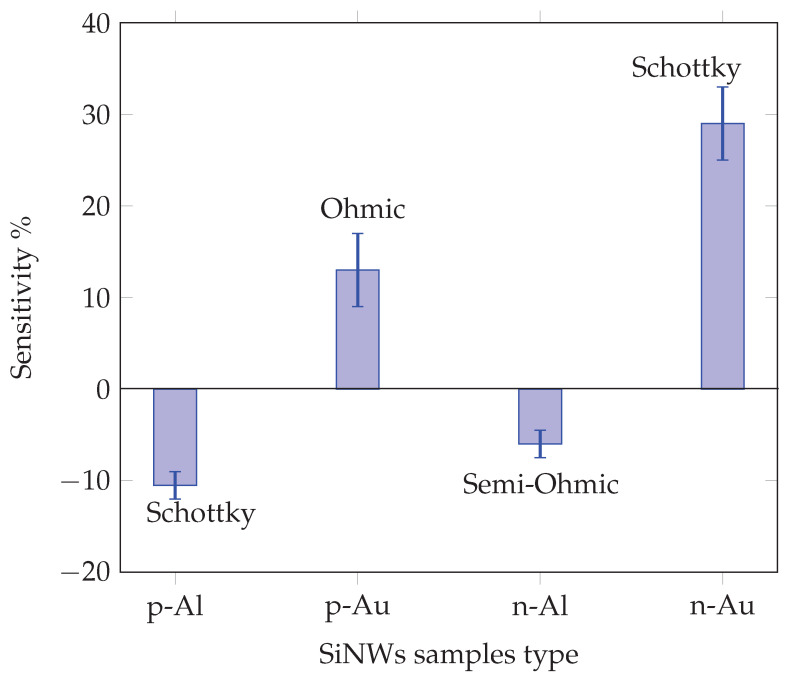
Sensitivity of SiNWs samples, error bars shows the range of 10 individual measurements.

**Figure 10 sensors-23-09901-f010:**
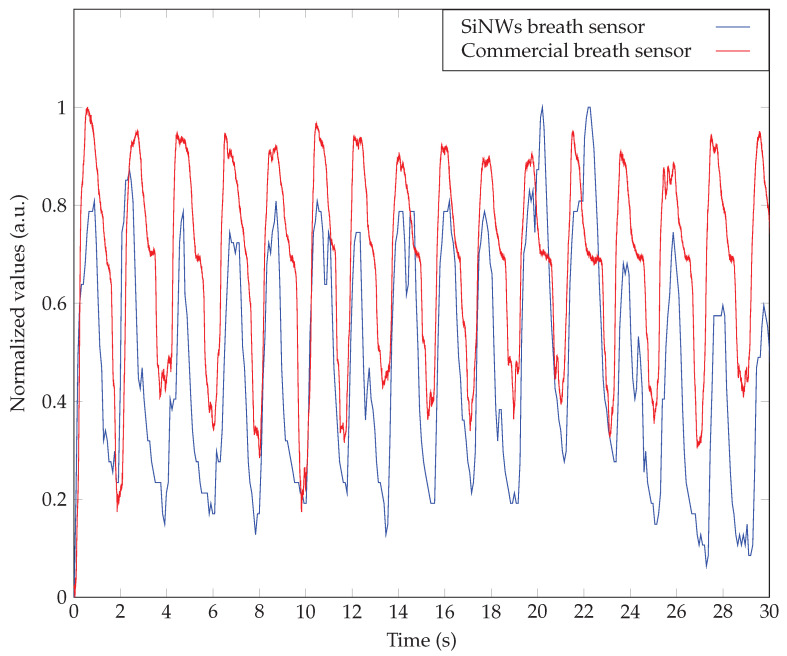
Comparison between a commercial breath sensor (Nox A1s™) and a SiNWs breath sensor.

**Figure 11 sensors-23-09901-f011:**
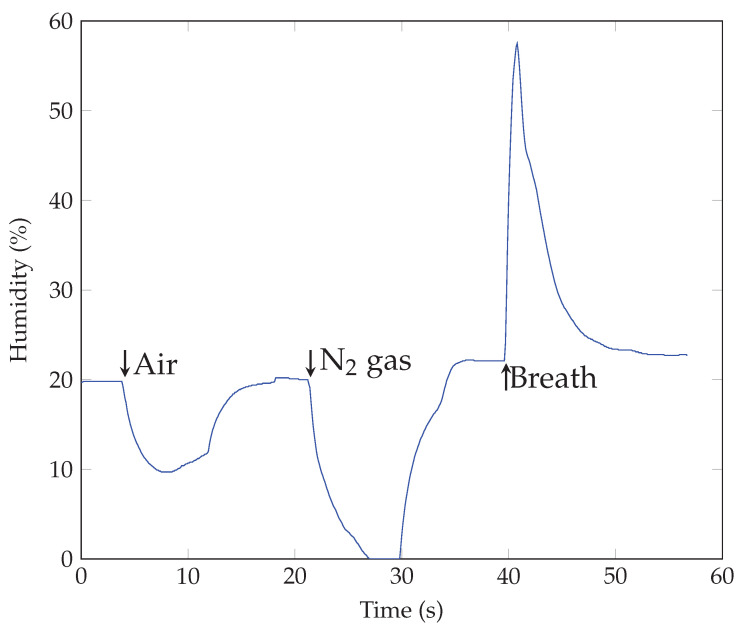
Humidity level changes, measured by a commercial humidity sensor.

**Figure 12 sensors-23-09901-f012:**
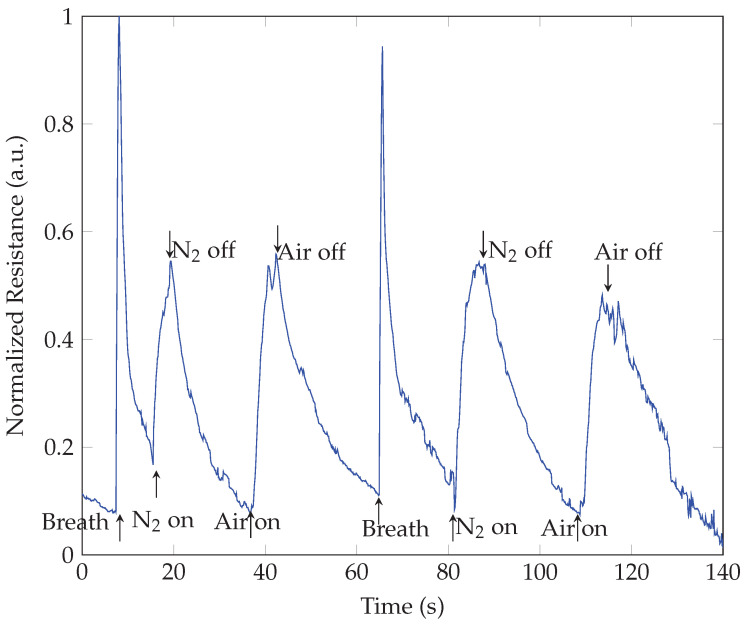
Resistance changes on p-type SiNWs sample with Au electrodes upon exposure to breath, N${}_{2}$-gas, and compressed air.

**Table 1 sensors-23-09901-t001:** List of fabricated and tested samples with measured I–V characteristics.

Electrodes	p-Type SiNWs	n-Type SiNWs
Au	Ohmic	Schottky
Al	Schottky	Semi-ohmic

## Data Availability

Data are contained within the article.

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
