# Peer review of "Application of p and n-Type Silicon Nanowires as Human Respiratory Sensing Device"

_sensors, 2023, doi:10.3390/s23249901_

Round 1

Reviewer 1 Report (Previous Reviewer 3)

Comments and Suggestions for Authors

The authors have made substantial efforts to address all reviewers’ comments, and the revised manuscript looks much better. The manuscript could be accepted.

  Comments on the Quality of English Language

The quality of English Language is fine.

Author Response

Dear reviewer

We appreciate your comments that helped to improve our manuscript.

Best regards

Elham Fakhri

Reviewer 2 Report (Previous Reviewer 4)

Comments and Suggestions for Authors

The manuscript is now better than the first time, but it still requires some mandatory changes. The authors only chose to address some of my earlier comments, whereas in fact they should address them all. I repeat my comments from last time that were not addressed:

"The significance of the work, in the context of previous work reported in literature, is not well presented. Other novel materials that are used for real time respiration monitoring, such as optical fibres, polymers, and graphene, are only briefly mentioned at the end of the paper, and in a rather negative context which is unfair and unrealistic. Most of the solutions that have already been reported, from other materials, are neither complicated to make, nor unstable in time, as the authors suggest. To name a few, competing technologies are fiber-optic meta-tips, Er2O3 nanospheres, Langmuir-Blodgett graphene films, nano-structured electrochemically active aluminum, and plasma-modified graphene."

Hence, the paper still lacks an honest comparison to earlier work that was performed with other materials, which yielded sensors with similar performance to the one shown here. It is not fair to mention just this one material and pretend it's a unique solution.

Author Response

Dear reviewer

Thank you for your comments, the suggestions have been added to the text in red color in the manuscript.

Best regards

Elham Fakhri

Reviewer 3 Report (Previous Reviewer 6)

Comments and Suggestions for Authors

The authors have largely revised their manuscript, it is acceptable for publication.

Author Response

Dear reviewer

We appreciate your comments that helped to improve our manuscript.

Best regards

Elham Fakhri

Round 2

Reviewer 2 Report (Previous Reviewer 4)

Comments and Suggestions for Authors

The authors have somewhat improved the manuscript by adding several references to other materials as humidity sensors.

This manuscript is a resubmission of an earlier submission. The following is a list of the peer review reports and author responses from that submission.

Round 1

Reviewer 1 Report

Comments and Suggestions for Authors

In this article, the author employed metal-assisted chemical etching to fabricate a sensor based on SiNWs for real-time breath detection. The article compares the detection performance of sensors created using p-type and n-type materials, along with those integrated with Au or Al attachments. While the article is well-written, certain results appear to be absent. A significant revision is recommended.

1. When introducing a chemical element, like on Line 43 where the authors employed "Au," and on Line 35 where it is denoted as "Gold," a consistent practice is to initially use the full chemical name. This pattern applies similarly to Aluminum (line 53) presented as "Aluminum," and then abbreviated as "Al" on Line 43.

2. In their earlier research, the authors demonstrated the heightened sensitivity achieved by incorporating Ge nanoparticles onto SiNWs. Could the current design in this manuscript potentially yield amplified sensitivity with the integration of Ge nanoparticles?

3. It is suggested to redraw Figure 1 (PCB board) with 3D representation. I believe it will be clearer for readers. 

4. Can the author provide the repeatability of the SiNWs sensor? Are there considerable deviations observed among experiments?

5. Nanostructures typically exhibit variations between each fabrication batch. Electrochemical sensing is recognized for its sensitivity. I wonder whether the authors encountered any discrepancies in detection results when comparing nanostructures fabricated from different batches.

Reviewer 2 Report

Comments and Suggestions for Authors

Comment and observations  

Dear authors, thank you for this very interesting paper. Application of p and n-type silicon nanowires as human respiratory sensing device is very interesting the work. There are some recommendation to improve the quality of this contribution. Thank you for sharing this work and knowledge.

Recommendation: I recommend that the article be accepted but the authors have to make deep corrections, otherwise the article is rejected.   Here are my comments:  

1.      It is necessary that all the manuscript must be revised and corrected because there are grammatical errors.

2.      The authors say “In an experimental study by Novikov et al., [16] the dependence of the work-function of silicon to doping concentration was shown to be in the range of only  +-0.03 eV for an order of magnitude different doping concentration. As in our study, where the resistivity of the n-type SiNWs samples is an order of magnitude lower than that of the p-type SiNWs samples, the work-function is expected to be just slightly different for both samples.”  How do the authors verify that the value of the work function of the silicon of the nanowires are similar? It is mandatory to make an experimental measurement to obtain the work function, since this is a fundamental part of the study of this investigation.

3.      How many samples were manufactured and measured to get the I-V curve according each sample?  It is mandatory to specify the average of each sample? Include this information in the manuscript.

4.      The contact metal-SiNWs in general when both surface are in contact, after manufacture of the layers, always it is recommendable to apply thermal annealing to be sure that exist a good metallurgical contact and so you can sure that the Schottky or ohmic contact is present. So that, Do you apply some thermal annealing to devices?  What was the conditions? This information should be included in the manuscript.

5.      In this sentence the authors say “As seen in Figs. 4 and 5, the sensor’s resistance increases with exhaling and decreases with inhaling for both n and p-types samples with Au electrodes (fSi < fAu)” . First, from figures 4 and 5 it is not clear the cycle when start the inhaling and exhaling process. Please indicates with a diagram this part to identify both process. Second, you says that the resistance increases with exhaling and decreases with inhaling for both types of materials, but if you analyze the resistance for both devices it is not stable, what is the reason of this behavior?

In this part, it is well know that the resistance nature, always is affected by the temperature, so that in your analysis and devices these are affect by this variable and you not yet included and it is important. Therefore, you need to do temperature compensation to get a stable device. Also, the changes of resistance in your devices clearly are been affected by the type material that you use. When you start the experiments with the devices, you must be sure that the temperature is controlled. After it is necessary to do variations with the temperature.   Please, it is necessary to include this experimental part to discard anything behavior on the measurements and so you have enough arguments to explain the phenomena.

If you show this results without the temperature behavior, these are not enough to support the investigation, because you can get any results and therefore, there are not repeatability.  

6.      Regarding the comparison when you use a commercial humidity sensor as is shown in Fig. 8, it is necessary to do the same comparison with your sensors. Under this way you can compare both results regarding the reference. Please, it is mandatory to do this analysis and experiments with humidity. Include in the manuscript.   

7.      From Fig. 9 Resistance changes on p-type SiNWs sample with Au electrodes upon exposure to breath, N2-gas and compressed-air. What is the behavior with the others sensors? In the same Fig. 9 when the device is exposed to N2 and air, the reading difference between them is almost the same. What and why is the reason? Give more arguments and forceful. Include in the manuscript.

8.      In conclusion you say “Other variables, such as due temperature changes, may also affect the results”, In all the manuscript you do not report nothing about the temperature or experimental part, Why is the reason of this sentence?

9.      What is the sensibility of this sensor? Must be included this analysis in the manuscript.

10.    The output voltage of this sensor, what is the value?

11.   The study of the piezoresistive effect and the gauge factor for this material must be included in this investigation to give more support to results. The authors must include some analytical expression or simulation to support the results and these be shown in the manuscript.

12.   Regarding the experimental part and fabrication of the sensor. How many sensor were manufactured and measured?  It necessary include a statistical study to be sure of that this behavior is repetitive for all the sensor? Include this analysis in the paper.

Comments on the Quality of English Language

Moderate editing of English language required

Reviewer 3 Report

Comments and Suggestions for Authors

In this paper, a human respiratory sensor based on SiNWs is presented. The results are acceptable, and the topic can attract a wide range of readers. However, there are several issues in the introduction, presentation, and discussion of the results. Therefore, a major revision is necessary before considering publication. My specific comments are as follows:

  1. Abstract: The authors mentioned the piezoresistance effect as the cause of resistance changes, but they should also investigate the influence of temperature and humidity on the sensor’s performance.

  2. Introduction and discussion: The manuscript lacks clarity regarding its logic, motivation, and innovation. The authors should improve the following aspects: (1) explain the reason for material selection with more clarity, (2) enhance the presentation of the innovative aspects, and (3) review and analyze recent research on advanced respiratory sensors to emphasize the uniqueness and innovation of this work.

  3. Figure 1: The authors should provide convincing evidence, such as a performance demonstration video.

  4. The authors should provide performance parameters of the sensor, including sensitivity, response/recovery times, detection range, and hysteresis.

  5. It is recommended to summarize the performance parameters of different human respiratory sensors in a table, including sensitivity, response/recovery times, and limit of detection.

  6. As a research paper, the sensing mechanism requires a systematic analysis.

  7. Figure 9: The authors should provide the actual resistance value.

  8. The quality of the figures needs improvement.

  9. References list: Most of the references are outdated. It is suggested that references should primarily focus on the past three years.

  10. Carefully review the full text and eliminate writing errors. For example, on page 3, Line 91, there should be a space between the numerical value and the unit.

Comments on the Quality of English Language

The English writing requires further polishing.

Reviewer 4 Report

Comments and Suggestions for Authors

The paper is superficial, with no strong scientific conclusion. All the conclusions made by the authors are very weak, for example in the abstract: “The mechanism behind this effect seems complex”. Even the authors themselves admit that the work is mostly qualitative.

The significance of the work, in the context of previous work reported in literature, is not well presented. Other novel materials that are used for real time respiration monitoring, such as optical fibres, polymers, and graphene, are only briefly mentioned at the end of the paper, and in a rather negative context which is unfair and unrealistic. Most of the solutions that have already been reported, from other materials, are neither complicated to make, nor unstable in time, as the authors suggest. To name a few, competing technologies are fiber-optic meta-tips, Er2O3 nanospheres, Langmuir-Blodgett graphene films, nano-structured electrochemically active aluminum, and plasma-modified graphene.

The authors' effort to find out the mechanism behind the resistance changes during breathing is rudimentary. It is hypothesized that both humidity and pressure can cause the observed changes, and an effort is made to test the two independently. However, pumping air and breathing on the devices will change both pressure and humidity at the same time. It would have been more appropriate if pure mechanical pressure was applied on the devices to decouple from changes due to humidity.

Comments on the Quality of English Language

The English has much room for improvement. Aside from grammatical errors, the language is often colloquial, ex. “work-functions whose levels are on different sides of silicon´s work function”, instead of “work functions lower and higher than the WF of silicon, respectively”. The use of colloquial language adds to the feeling that the work was not performed with maximal care and effort.

Reviewer 5 Report

Comments and Suggestions for Authors

1) Its quite difficult to understand the fabrication process of the device with just several paragraphs of texts. A schematic would be helpful to understand the methods in section 2.1 lines 60-78.

2) How does the performance of the sensor presented in this paper compared to the state of the art? What limitations does the author's device have that need to be overcome to make this device successful?

3)Do the operating conditions of the device change its performance? i.e. Does the weather/ambient humidity and temperature affect the performance of the device?

Comments on the Quality of English Language

N/A

Reviewer 6 Report

Comments and Suggestions for Authors

The authors reported a human respiratory sensing device using p and n-type silicon nanowires. The sensors were fabricated using a top-down method of metal-assisted-chemical-etching and exhibited the ability to monitor the human breath condition. This work is interesting, but some major issues need to be addressed before considered for publication.

1.         What is the main innovation of this work compared with the other SiNW-based respiration sensors? A brief discussion should be added.

2.         The performance characterizations of sensors are mainly using human breath, which is insufficient for the accuracy measurement of the device. For example, to evaluate the sensor’s humidity response, the devices should be measured in a humidity chamber, and record the resistance change according to the humidity change.

3.         For the experiment in Figure 9, the pressure of air, N2, and breath should be accurately controlled as the same pressure, how did the authors achieve such a condition? Directly measuring the humidity response of the sensor in a humidity chamber is better than the present method.

4.         How do the authors characterize the response time of the sensor? The measurement method and the result should be provided.

Comments on the Quality of English Language

The clarity of discussion needs to be improved